# Practical access to axially chiral sulfonamides and biaryl amino phenols via organocatalytic atroposelective *N*-alkylation

Shenci Lu [1,2], Shawn Voon Hwee Ng [2], Kaitlyn Lovato [3], Jun-Yang Ong[2], Si Bei Poh[2], Xiao Qian Ng [2], László Kürti[3] & Yu Zhao [2]

The importance of axial chirality in enantioselective synthesis has been widely recognized for decades. The practical access to certain structures such as biaryl amino phenols known as NOBINs in enantiopure form, however, still remains a challenge. In drug delivery, the incorporation of axially chiral molecules in systematic screening has also received a great deal of interest in recent years, which calls for innovation and practical synthesis of structurally different axially chiral entities. Herein we present an operationally simple catalytic *N*-alkylation of sulfonamides using commercially available chiral amine catalysts to deliver two important classes of axially chiral compounds: structurally diverse NOBIN analogs as well as axially chiral *N*-aryl sulfonamides in excellent enantiopurity. Structurally related chiral sulfonamide has shown great potential in drug molecules but enantioselective synthesis of them has never been accomplished before. The practical catalytic procedures of our methods also bode well for their wide application in enantioselective synthesis.

[1] Shaanxi Institute of Flexible Electronics, Northwestern Polytechnical University, Xi'an 710072, China. [2] Department of Chemistry, National University of Singapore, 3 Science Drive 3, Singapore 117543, Singapore. [3] Department of Chemistry, Rice University BioScience Research Collaborative, 6500 Main Street, Rm 380, Houston, TX 77030, USA. Correspondence and requests for materials should be addressed to L.K. (email: kurti.laszlo@rice.edu) or to Y.Z. (email: zhaoyu@nus.edu.sg)

Molecular chirality plays an essential role in drug delivery as enantiomeric molecules perform distinct interactions with homochiral macromolecules in biological systems[1,2]. In contrast to the ubiquitous central chirality, axial chirality stems from hindered rotation around a bond axis, which was unfortunately overlooked for many years in drug delivery due to the dynamic nature of such chiral compounds. With the support of much recent synthetic efforts[3–5], the significance of axially chiral compounds in systematic drug screening has attracted a great deal of attention, and are considered to possess unique advantage in delivering stable and compact structures for small molecule drugs[6,7]. Out of the different classes of axially chiral compounds, biaryls bearing hindered rotation represent the most recognized type[8–10]. Molecules such as knipholone (Fig. 1a) have found much use in traditional medicine. In addition, the discovery of new bioactive biaryl compounds also represents a highly active area of research. In the realm of enantioselective catalysis, axially chiral biaryls have proven to be privileged structures as well[11]. In particular, 1,1'-bi-2-naphthol (BINOL) and related compounds have found extensive use as catalysts in a myriad of stereoselective transformations[12,13]. A wide range of enantiopure BINOL analogs have been made readily available and are routinely screened for the development of new enantioselective transformations. The accessibility of a large library of them has proven to be essential for reaction optimization.

Despite the great advances in this area of research, significant limitations still exist for the practical access to certain important structures, such as the axially chiral amino alcohol NOBIN that has shown great utility in asymmetric catalysis (Fig. 1b)[14]. In fact, the preparation of NOBIN in enantiomerically pure form still represents a significant challenge. Although many attempts have been documented on traditional oxidative coupling of two aryl fragments[15], classical resolution[16], as well as catalytic kinetic resolution[17,18] and enantioselective synthesis[19], most of the known methods possess limitations such as low efficiency (the use of stoichiometric chiral reagents) or the use of catalysts that take multiple steps to prepare. In fact, due to their cumbersome synthesis, NOBIN analogs bearing substituted chiral backbones have rarely been prepared and explored[14]. Therefore, a general, practical and scalable synthetic approach to enantiopure NOBIN and analogs is still highly desired.

The phenomenon of axially chirality is not limited to hindered biaryls. Much recent efforts have also been devoted to the identification of new axially chiral compounds as promising structural motifs in pharmaceuticals. Tertiary amides and anilides bearing a chiral axis (Fig. 1c), in particular, have attracted much attention in recent years and many efficient catalytic systems have been developed for their enantioselective synthesis[20–25]. Among the different strategies, the construction of the key amide bond or the aryl-carbonyl C–C bond represents the most commonly explored approach. In sharp contrast, substituted sulfonamides are largely uncharted for axially chiral entities, despite the fact that some molecules containing this structural motif have been recognized as potent drug candidates in the pharmaceutical industry. One example is UK-240455, which is an effective NMDA antagonist[26]. Surprisingly, the enantioselective synthesis of this class of axially chiral compounds remains elusive in the literature and represents a significant gap in the synthetic method development.

In this paper, we present our recent efforts in the development of a unified and practical catalytic procedure to deliver two important classes of axially chiral molecules (i.e., NOBIN analogs and chiral sulfonamides) in high efficiency and excellent enantiopurity (Fig. 1d). The first transformation provides an effective kinetic resolution of NOBIN analogues, which can serve as valuable chiral catalyst precursors. The second process converts readily available aniline-derived sulfonamides to axially chiral sulfonamides in excellent yield and enantioselectivity. This method is practical, utilizes readily available reagents/catalysts and can be easily scaled up with straightforward recovery of the chiral catalyst. Derivatization and application of the chiral compounds accessed using this method have also been demonstrated.

## Results

**Access to enantiopure NOBIN analogs by *N*-alkylative kinetic resolution.** Kinetic resolution is a highly effective strategy to produce a general library of enantiopure NOBIN analogs. Despite the inherent 50% yield limitation, the conversion of a resolution process can be controlled to reliably access the unreacted substrates in a nearly enantiopure form (≥98% ee)[27,28]. The availability of the racemic substrates, on the other hand, largely influences the effectiveness of a resolution process. Our previous work resulted in a highly practical and scalable organocatalytic preparation of a wide range of racemic biaryl sulfonamide phenols in high yields (see below)[29,30]. If this racemic synthesis can be combined with a catalytic kinetic resolution using a commercial and inexpensive catalyst, this sequence may provide a practical access to enantiopure NOBIN library. Bearing this in mind, we initiated our investigation by exploring different catalytic strategies for the resolution of **1a**. These strategies included an NHC-catalyzed acylation of the phenolic OH group, which was done in our previous work[18,31], and an amine-catalyzed substitution of the sulfonamide N–H bond using various reagents.

From these attempts, *N*-alkylation of **1a** using the MBH carbonate **2a** under chiral amine catalysis[32] proved to be the most promising. Moderate conversion and selectivity were obtained for the alkylation using commercial quinine **A** as the catalyst (Table 1, entry 1). Encouraged by this initial lead, we examined a series of commercially available chiral amines, including monomeric and dimeric cinchona alkaloid-derived compounds (Table 1, entries 2–6). (DHQD)$_2$AQN **E** proved to be the optimal catalyst for this reaction, resulting in good selectivity ($S = 9$) and moderate efficiency (Table 1, entry 5).

Other reaction parameters such as the nature of the solvent, catalyst loading, and steric bulk of the alkylating agent were evaluated next. The choice of solvent showed to be particularly influential (Table 1, entries 7–9). The use of CH$_3$CN led to a much-improved reactivity, although with a low selectivity ($S = 4$). When we examined the solvent combination of CH$_2$Cl$_2$ and CH$_3$CN, a good balance of yield and selectivity was achieved (Table 1, entry 10). Employing a bulkier MBH carbonate **2b**, which can be prepared in one step, led to a significantly improved selectivity of $S = 21$ (Table 1, entry 11). Importantly, the reduction in the catalyst loading did not affect the enantioselectivity of this process (Table 1, entry 12). However, a higher loading of catalyst was needed to achieve good conversion and in turn an excellent ee for the recovered (*R*)-**1a**. It is important to note that the catalyst can be easily recovered from this process.

With the optimal conditions in hand, we examined the scope of this catalytic kinetic resolution (Fig. 2). All the racemic substrates, in which a polysubstituted aryl group bears the sulfonamide moiety, were easily accessed by our efficient organocatalytic procedure. The S factor was determined based on the ee of the product and recovered substrate. The model substrate **1a** was recovered in 38% yield with 98% ee. Having a Br substituent in the 3-position led to a very high selectivity and enantiopure **1b** (99% ee) was recovered in 40% isolated yield. Notably, for substrate **1c** bearing a 3-methoxy group, the selectivity dropped to $S = 11$. However, the substrate could still be recovered with a high 95% ee and only a slightly lower yield of 32%. This showcased the flexibility of a kinetic resolution process and its ability to be used for library synthesis.

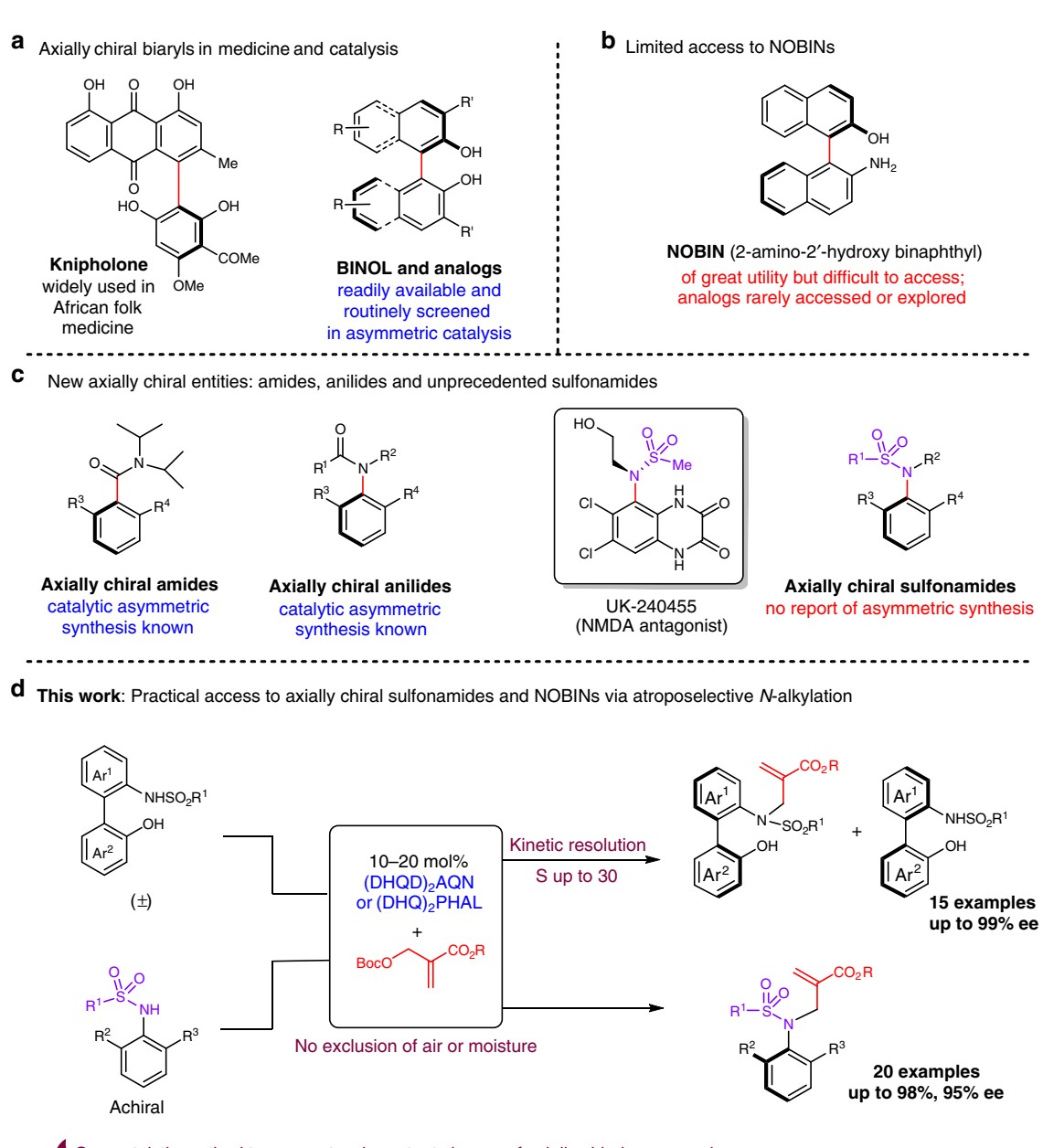

**Fig. 1** Application and preparation of various axially chiral entities. **a** Representative well-established chiral biaryls. **b** Axially chiral amino alcohol NOBIN. **c** Other types of axially chiral entities. **d** Our strategies to axially chiral sulfonamides and NOBINs. BINOL: 1,1′-bi-2-naphthol

Examples **1d–1g** exemplified that substrates with a variety of functional groups (i.e., halogens, esters, and ethers) in different positions, could also produce good to high level of enantioselectivity for the *N*-alkylation reactions. These substrates were recovered in excellent enantiopurity (95–99% ee). Similar to the Ms-protected anilines, the tosyl-protected **1h** could undergo the resolution reaction with the same level of enantioselectivity. Amino phenol **1i**, which features a more bulky fused aryl group, was also recovered in an excellent 99% ee. However, in the case of other halogenated substrates such as **1j** and **1k**, the lower efficiency of *N*-alkylation led to an overall lower enantiopurity. Nonetheless, further optimizing the conditions to improve the reaction conversion should allow these compounds to be recovered in excellent enantioselectivity.

To further explore the scope of this catalytic method beyond the NOBIN analogs available from our previous synthesis[30], other NOBIN analogs bearing different anilines such as **1l**[29] or binaphthyl-derived NOBINs including **1m–1o** were prepared using alternative syntheses. These analogs were subjected to the kinetic resolution and gratifyingly, similar or even higher levels of reactivity and enantioselectivity could be achieved. Our catalytic kinetic resolution can thus deliver a wide range of NOBIN analogs with biphenyl, binaphthyl, as well as phenyl–naphthyl backbones in excellent enantiopurity. Such diversity is of great value for reaction development. It is noteworthy that the Tan group has reported an elegant and direct chiral phosphoric acid-catalyzed preparation of NOBIN analogs in high yield and excellent enantioselectivity[19]. However, in order to obtain the

**Table 1 Optimization of resolution of amino phenol 1ᵃ**

| Entry | 2 | Cat | Solvent | Conv. (%)$^b$ | Product, ee (%)$^c$ | 1a, ee (%)$^c$ | $S^d$ |
|---|---|---|---|---|---|---|---|
| 1 | 2a | A | CH$_2$Cl$_2$ | 30 | 3a, 60 | 26 | 5 |
| 2 | 2a | B | CH$_2$Cl$_2$ | 26 | 3a, −26 | −9 | 2 |
| 3 | 2a | C | CH$_2$Cl$_2$ | 76 | 3a, −7 | −23 | 1.4 |
| 4$^e$ | 2a | D | CH$_2$Cl$_2$ | 5 | 3a, 73 | 4 | 6.6 |
| 5 | 2a | E | CH$_2$Cl$_2$ | 28 | 3a, 74 | 29 | 9 |
| 6$^e$ | 2a | F | CH$_2$Cl$_2$ | 9 | 3a, 78 | 8 | 8.7 |
| 7 | 2a | E | CH$_3$CN | 80 | 3a, 22 | 90 | 4 |
| 8 | 2a | E | THF | 28 | 3a, 20 | 8 | 1.6 |
| 9 | 2a | E | EtOAc | 28 | 3a, 36 | 14 | 2.4 |
| 10 | 2a | E | 1:1 CH$_2$Cl$_2$/CH$_3$CN | 62 | 3a, 50 | 82 | 7 |
| 11$^f$ | 2b | E | 1:1 CH$_2$Cl$_2$/CH$_3$CN | 60 | 4a, 65 | 98 | 21 |
| 12$^g$ | 2b | E | 1:1 CH$_2$Cl$_2$/CH$_3$CN | 39 | 4a, 86 | 55 | 23 |

$^a$Unless noted otherwise, the reactions were performed with **1a** (0.04 mmol, 1.0 equiv.), **2** (0.8 equiv.), catalyst (20 mol%) in solvent (0.5 mL) at 24 °C for 4 h
$^b$Determined by $^1$H NMR
$^c$Determined by chiral HPLC
$^d$$S = \ln[(1-\text{Conv.})(1-ee_{1a})]/\ln[(1-\text{Conv.})(1+ee_{1a})]$
$^e$15 h instead of 4 h
$^f$The reaction time was 24 h
$^g$Use of 10 mol% **E** and with 24 h reaction time

products in >90% ee an aryl protecting group on the NOBIN nitrogen is required.

The impact of substitution on the amino and hydroxyl groups was also examined. In contrast to the $N$-Ms substituted **1a** that was produced with a high enantioselectivity ($S = 21$), the resolution of a $N$-tert-butoxycarbonyl (i.e., Boc-carbamate) substrate **1p** led to an enhanced reactivity but with a dramatic loss in selectivity ($S < 2$). This highlights the importance of the sulfonamide moiety in enantioselectivity control in this $N$-alkylation reaction. On the other hand, when the phenol moiety was protected as a methyl ether, a complete loss of enantioselectivity was observed for the kinetic resolution of **1q**. Although the exact nature of this effect has not been confirmed, it is likely that the free phenol may interact with the basic moiety on the catalyst, which could rigidify the transition state structure and thus induce high enantioselectivity[31].

To further showcase the practicality of this catalytic system, the gram-scale preparation of the substrate **1b** followed by the

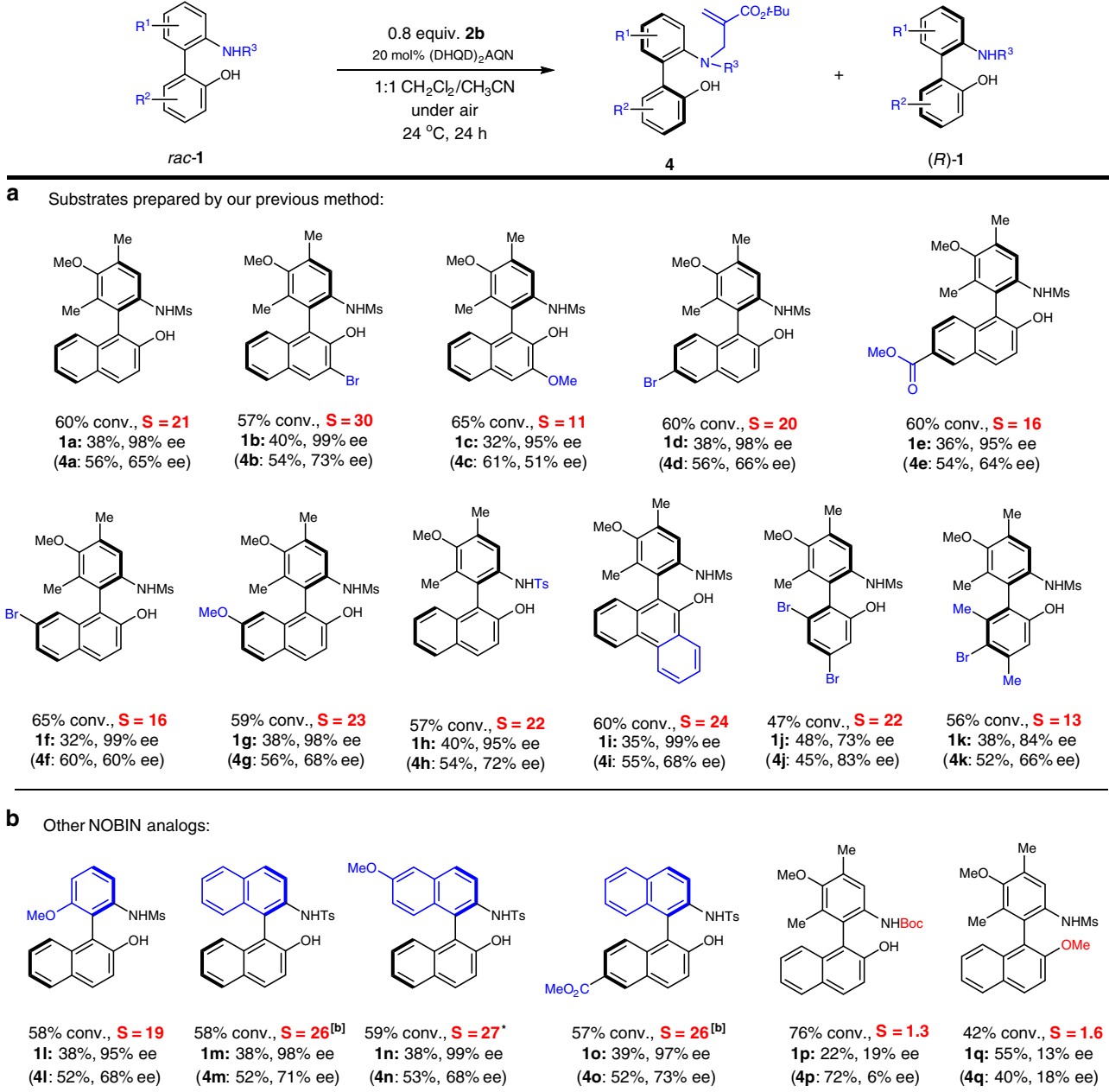

**Fig. 2** Scope and structural exploration of NOBINs from kinetic resolution. **a** Substrates from one-step biaryl sulfonamide phenol synthesis. **b** More diverse biaryl amino phenols. *0 ℃ reaction for 48 h

kinetic resolution of **1b** using a reduced catalyst loading (10 mol %) was carried out (Fig. 3a). Enantiopure (R)-**1b** was obtained in 35% yield. Single crystal X-ray analysis of **4b** further confirmed the structure of this N-alkylated product. In addition, the catalyst was recovered in 90% yield. When the recovered catalyst was used for another round of kinetic resolution, the recovered **1b** could be obtained with the same efficiency and enantioselectivity.

In an effort to recycle the sacrificial product in this kinetic resolution, product **4b** was subjected to Ni-catalyzed de-allylation to regenerate **1b** in 95% yield with 60% ee (Fig. 3b). The use of another chiral amine catalyst, (DHQ)₂PHAL, resulted in an enhancement of the enantiopurity of (S)-**1b** to 97% ee. In this way, both enantiomers of the axially chiral amino phenol **1b** could be accessed.

The enantiopure halogen-containing amino phenols can be derivatized through Pd-catalyzed cross coupling, which further expands the NOBIN library. As shown in Fig. 3c, coupling of (R)-**1b** with phenyl boronic acid furnished **5a** bearing different substituents ortho to the phenol moiety in high yield without optimization. Similarly coupling of **1f** produced **5b** with an altered backbone in 82% yield. Based on our previous report[30], the removal of the Ms group could also be carried out to deliver free amino phenol **6a** in high yield. This free amino phenol could be further modified using the same Pd-catalyzed cross coupling to yield **6b** in a good yield.

The method provides highly efficient access to enantiopure axially chiral amino alcohols bearing a polysubstituted aniline moiety. We set out to explore the utility of this class of NOBIN analogs in asymmetric catalysis. As shown in Fig. 3d, pyridine-

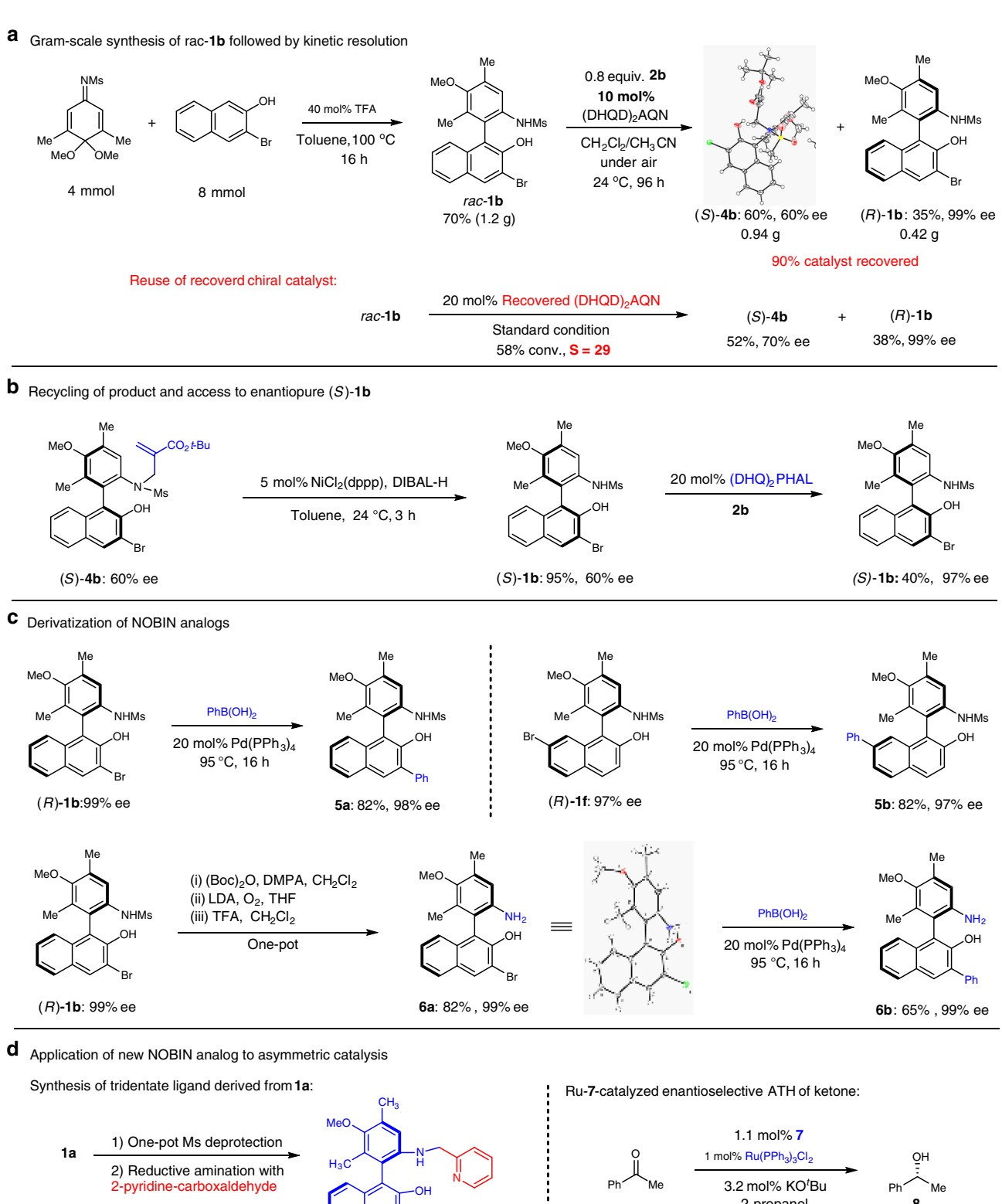

**Fig. 3** Large scale synthesis, derivatization and application of NOBIN analogs. **a** Gram-scale substrate synthesis and resolution. **b** Access the the enantiomeric NOBIN analog. **c** Further derivatization to enlarge the scope of NOBIN analogs. **d** Application to asymmetric catalysis

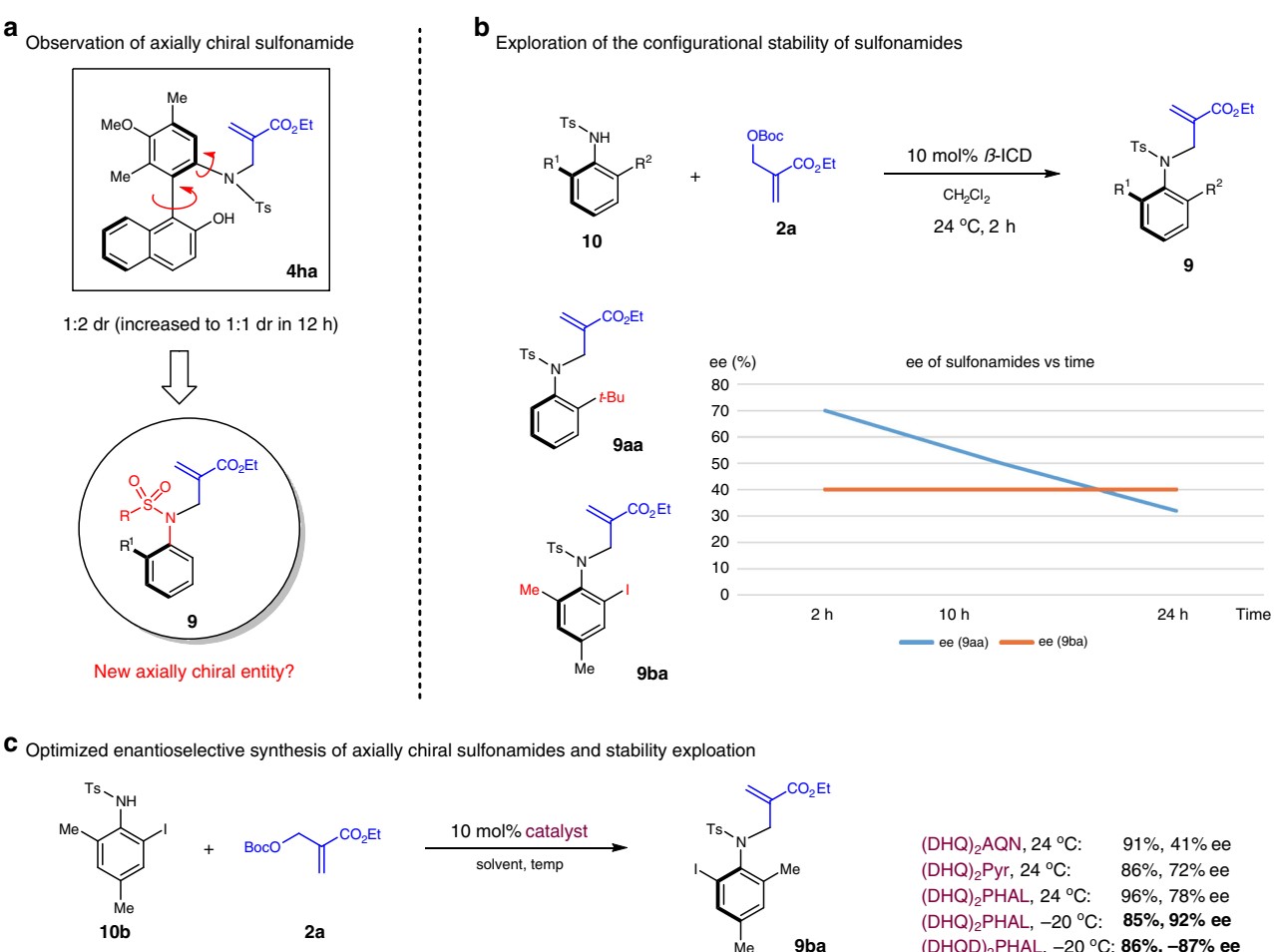

**Fig. 4** Discovery and exploration of axially chiral sulfonamides. **a** Observation of axial chirality in N-aryl sulfonamide. **b** Exploration of the stability of axial chirality. **c** Optimization of atroposelective sulfonamide synthesis

containing amino alcohol **7** could be easily accessed from **1a** using two one-pot procedures (see Supplementary Information for details). When amino alcohol **7** was used as a chiral ligand with standard Ru-catalyzed asymmetric transfer hydrogenation conditions (i.e., conditions which typically utilize the parent NOBIN-derived ligand[33]), a high yield of 90% was obtained for alcohol **8** with an excellent 93% ee. This serves as a convincing proof-of-principle for the utility of this class of axially chiral compounds.

**Access to chiral sulfonamides by enantioselective N-alkylation.** An unexpected discovery was made during our optimization of the formation of **4ha** (Fig. 4a). Under certain reaction conditions, a mixture of two isomers was observed for the formation of the tertiary sulfonamide **4ha** (see Supplementary Information for details). In addition, the ratio of the two isomers changed over an extended period of time. We were intrigued that this observation should be attributed to the possibility of atropisomerism of the sulfonamide moiety. Interestingly, no enantioselective preparation of such functionality has been reported in the literature. Taking that into consideration, we decided to explore the formation of axially chiral N-aryl sulphonamide **9** using the same N-alkylation reaction, and initiated our efforts by examining the configurational stability of sulfonamides **9** bearing different substitution patterns.

As illustrated in Fig. 4b, compound **9aa** bearing a mono-ortho-tert-butyl substituent on the aryl group could be obtained in up to 70% ee by amine-catalyzed N-alkylation in 2h;

however, the enantiopurity of this compound decreased rapidly with longer reaction time (only 30% ee after 24 h). In contrast, the ee of **9ba** bearing two ortho-substituents on the aryl group remained constant after an extended period of time (i.e., after 96 h). With these results, we decided to focus on optimizing the enantiopurity of sulfonamides such as **9ba** with both ortho-substituents. The examination of different catalysts, solvents and temperature was carried out systematically, the representative data from which was shown in Fig. 4c (see supinfo for more details). Under the optimcal conditions using (DHQD)₂PHAL as the catalyst and comercially available **2a** at −20 °C, **9ba** could be obtained with a high 86% yield with an excellent 92% ee. It is important to note that the use of the diastereomeric catalyst (DHQ)₂PHAL could deliver the enantiomeric **9ba** in a slightly reduced 87% ee. Thus, both enantiomers of **9ba** can be accessed with good enantioenrichement using this straightforward catalytic system.

With the optimal conditions in hand, the scope of this catalytic synthesis of axially chiral sulfonamides was explored. As shown in Fig. 5, the change of ester moiety resulted in uniformly high enantioselectivity for **9ba**–**9be**. Single crystal X-ray analysis of **9ba** also confirmed the absolute configuration of this class of compounds. Next, the incorporation of additional halide substituents on the aryl group was well-tolerated to produce **9ca**–**9ea** in 89–91% ee. All these examples possesses an iodo- and a methyl substituents at the ortho-positions. We were curious whether the size difference of these two substituents was essential

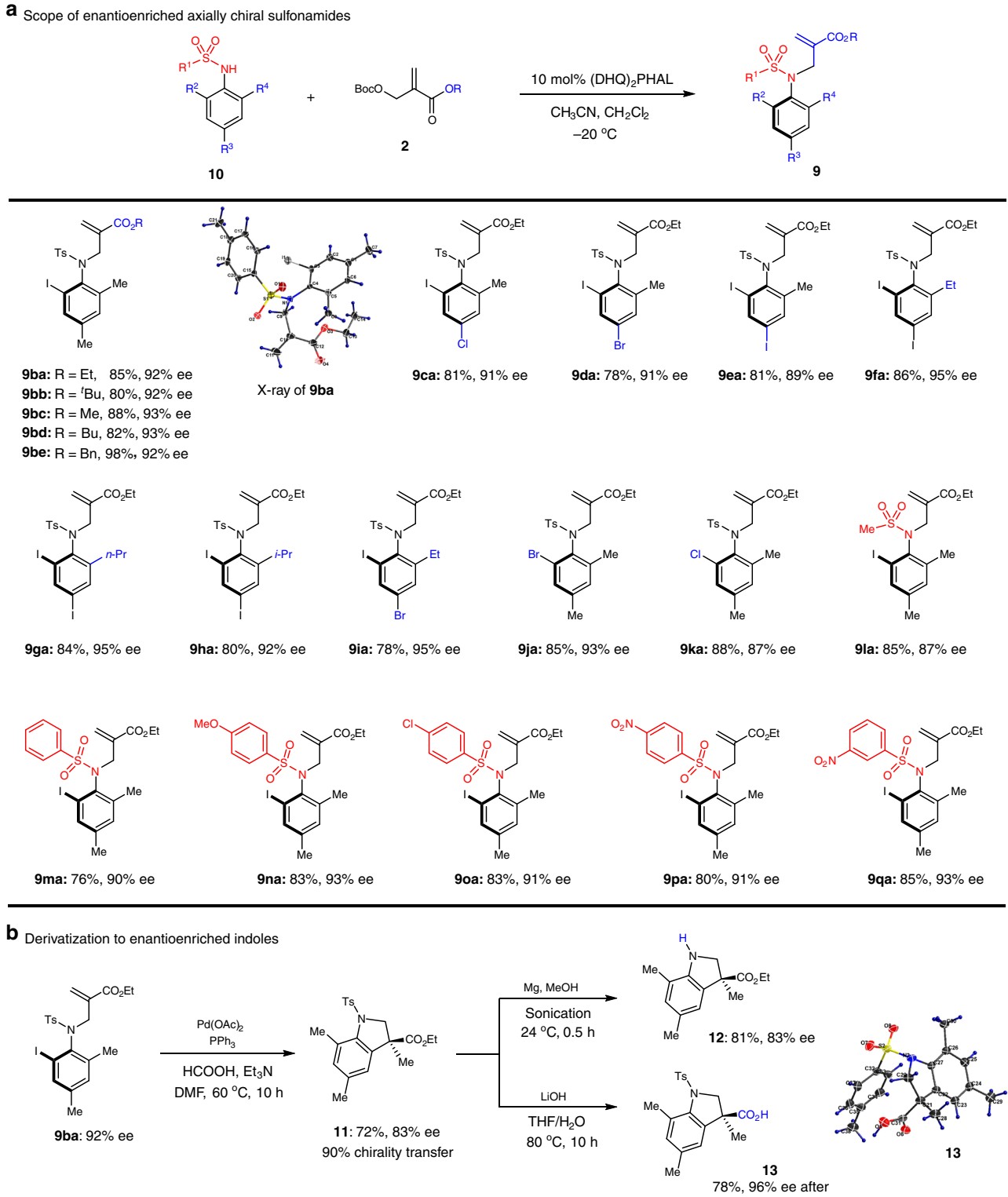

**Fig. 5** Scope of axially chiral sulfonamides and derivatization to enantioenriched indolines. **a** Scope of sulfonamides accessible using our method. **b** Access to indoles with retention of enantiopurity

for the high enantio-control. The test of substrates bearing larger alkyl groups including ethyl, propyl and even isopropyl was then carried out. As a pleasant surprise, products **9fa–9ha** were obtained in even higher enantioselectivities (92–95% ee). Even in the cases of **9ja** and **9ka**, which bears a bromo- or chloro-group vs. a methyl group, an excellent level of enantioselectivity could be obtained. We surmise that the presence of a halogen-substituent is key for this catalytic enantioselective system. Laslty, the variation of the sulfonamide moiety was examined. It was found that substrates from the small mesyl group to a range of substituted aryl solfonamidescould be synthesized with uniformly high ees (**9la–9qa**).

To further demonstrate the utility of this catalytic system, the derivatization of chiral sulfonamide **9ba** was carried out. Under palladium-catalyzed reductive Heck conditions, substituted indoline **11** could be obtained with good chirality transfer and high yield. It is worth noting that the enantioselective reductive Heck reaction still represents a challenge in synthetic chemistry[34]. In addition, the tosyl group could be easily removed to deliver the free indoline product **12** in high yield. Hydrolysis of **11** also delivered free acid **13**, whose crystal structure established the absolute configuration of these synthetically valuable indoline products.

## Discussion

We have developed a powerful and practical catalytic system that allows access to two classes of axially chiral compounds (i.e., structurally diverse NOBIN analogs and *N*-aryl sulfonamides) in high efficiency and enantioselectivity. The practical preparation of NOBIN analogs has been recognized as a serious limitation for decades. This catalytic system operates by *N*-alkylation using commercially available chiral amine catalyst and MBH carbonate reagent. These factors allow this method to be a practical option to access libraries of synthetically valuable NOBIN analogs for catalyst development. In addition, our method also provides an efficient synthesis of axially chiral *N*-aryl sulfonamides, a structural motif that has shown great potential in the pharmaceutical industry. The utility of this class of compounds in drug delivery is currently under investigation in our laboratories.

## Methods

**Representative procedure for the kinetic resolution of NOBIN 1.** To a 4 mL vial containing **1** (0.04 mmol) and (DHQD)$_2$AQN (7.0 mg, 20 mol%) were added CH$_2$Cl$_2$ (0.5 mL), CH$_3$CN (0.5 mL) and MBH carbonate (6 μL). The reaction mixture was allowed to stir at 24 °C for 24 h. The volatiles were removed *in vacuo* at 24 °C and the residue was purified by silica gel column chromatography with hexanes/ethyl acetate (10:1 v/v) as the eluent to afford the product **4** and unreacted starting material **1**.

**Representative procedure for synthesis of axially chiral sulfonamide 9.** To a 4 mL vial containing **10** (0.04 mmol) and (DHQ)$_2$PHAL (3.0 mg, 10 mol%) were added CH$_2$Cl$_2$ (0.25 mL) and CH$_3$CN (0.25 mL). The reaction mixture was allowed to stir for 10 min at −20 °C. Then MBH carbonate (12 μL) was added. Once the starting material **10** was consumed completely as judged by thin layer chromatography, the volatiles were removed in vacuo at 24 °C and the residue was purified by silica gel column chromatography with hexanes/ethyl acetate (10:1 v/v) as the eluent to afford the product **9**.

## Data availability

Experimental details, characterization of compounds, and copies of NMR data are available with the submitted manuscript. The X-ray crystallographic coordinates for structures reported in this study have been deposited at the Cambridge Crystallographic Data Centre (CCDC), under deposition numbers 1917938-1917941. These data can be obtained free of charge from The Cambridge Crystallographic Data Centre via www.ccdc. cam.ac.uk/data_request/cif.

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

## Acknowledgements

Y.Z. is grateful for the generous financial support from A*STAR SERC (R-143-000-648-305) and National University of Singapore (R-143-000-A57-114). L.K. gratefully acknowledges the generous financial support of Rice University, National Institutes of Health (R01 GM-114609-014) and the National Science Foundation (CAREER: SusChEM CHE-1546097) the Robert A. Welch Foundation (Grant C-1764), ACS-PRF (Grant 51707-DNI1), Amgen (2014 Young Investigators' Award for LK), and Biotage (2015 Young Principal Investigator Award) that are greatly appreciated. K.L. acknowledges the generous financial support of the National Science Foundation Graduate Research Fellowship (Grant no. DGE# 1842494). S.L. acknolwdges the financial support from the Fundamental Research Funds for the Central Universities (05150-19GH020157).

## Author contributions

S.L. and Y.Z. designed the project. S.L. carried out the experiments with S.V.H.N., K.L., J.-Y.O., S.B.P., and X.Q.N. L.K. and Y.Z. directed the project. S.L., L.K., and Y.Z. wrote the paper.

## Additional information

**Competing interests:** The authors declare no competing interests.

