## [Peer Review File · Nature Communications]

REVIEWERS' COMMENTS:

Reviewer #1 (Remarks to the Author):

In the submitted communication from the collaborative work by the Zhao and Kürti groups, a practical organocatalytic method for the kinetic resolution of NOBINs by an N-alkylation of sulfonamides is presented. An optimization of the reaction provided practical conditions for excellent selectivities at approx. 60% conversion, yields of approx. 40%/50% and selectivity factors of around 20. The absolute configuration was assigned by converting 1b and by analogy. The utility of the method was demonstrated by a gram scale reaction, recycling studies, derivatizations and an interesting application in a Rh-catalyzed ATH of a ketone. By increasing the size of the ortho-substituent of the aniline, restricted rotation about the C-N bond was observed and with o-disubstituted systems, sufficient configurational stability was achieved similar to previous reports with differently substituted anilides. Inspired by the work of Maruoka e.g. in Ref. 17, an enantioselective allylation of anilides was demonstrated and high selectivities were accomplished. The absolute configuration was determined by one X-ray measurement and by analogy. Derivatization to an enantioenriched indole furthermore shows the practicality of the method. The supporting information contains the required data and is of high quality. The manuscript is written clearly and the literature is cited appropriately. The only minor change to be suggested is to provide the configuration of indoles 11 and 12. In conclusion, this reviewer recommends acceptance for publication after the minor change mentioned.

Reviewer #2 (Remarks to the Author):

The manuscript by Kürti and Zhao reports a chiral amine-catalyzed atroposelective N-alkylation of sulfonamide substrates. This method leads to the resolution of NOBIN analogues and an enantioselective synthesis of axially chiral N-aryl sulfonamides.

The first part reports the resolution of NOBINs, which are important privileged structures in asymmetric synthesis. Despite the high catalyst loading and small S values, the current method still represents a practical and useful method to access important chiral NOBINs by resolution. Especially, the easy preparation of racemic NOBINs by the authors' reported method, and the ready availability of catalyst makes the current method more attractive.

The catalytic asymmetric synthesis of compounds possessing axially chiral Ar-N bond (e.g. axially chiral anilides) have been reported, however, the synthesis of analogue axially chiral N-aryl sulfonamides has not been reported. The second part of this manuscript reports the catalytic asymmetric synthesis of axially chiral N-aryl sulfonamides to address this issue.

Considering the importance of the products and the practicability of the method, this work might be suitable for publication in Nature Communications. However, the following issues should be addressed before acceptance of the manuscript.

(1) The current study used acid-catalyzed reaction to prepare racemic NOBINs, which were then used for resolution. However, the chiral acid-catalyzed asymmetric synthesis of NOBINs has been reported (Tan, B. et al. *Angew. Chem. Int. Ed.* 2017, 56, 16308), which seems to be more practical and attractive. In this case, the authors may need to clearly state the advantages of the current method over that report.

(2) The authors previously reported a kinetic resolution of NOBINs through NHC-catalyzed atroposelective acylation (*Angew. Chem. Int. Ed.* 2014, 53, 11041), and the strategy gave NOBINs with excellent enantioselectivities. The author mentioned they tried the previous conditions for the substrates in the current study, but no data was reported in the manuscript or in the SI. Is the previous strategy compatible with the current substrates?

(3) In the abstract, the authors claimed the importance of axially chiral N-aryl sulfonamides in 'drug delivery', however, the example provided by the authors in the main text is used as a NMDA antagonist, which seems to be a drug, rather than used for drug delivery.

(4) The structures of the amine catalysts in Table 1 are not in a consistent format. The authors should also check the stereocenters in the catalyst, e.g. the absolute configuration of chiral centers in (DHQD)2PHAL, (DHQD)2AQN, and (DHQD)2Pyr should be the same.

(5) In Scheme 5, what is the ee of compound 12? There is an optical rotation data for chiral 12 in the supporting information, so the author should also report the ee value and add the HPLC chart.

(6) The authors mentioned some of compounds 4 exist as a mixture of two isomers due to the presence of two axially chiral bonds. In this case, there should be four peaks for racemic 4 in the HPLC charts. However, there are only two peaks in all the HPLC charts of compounds 4. What's the reason?

Reviewer #3 (Remarks to the Author):

Zhao, Kürti, and coworkers report in this paper an organocatalytic atropenantioselective N-alkylation of sulfonamides for the synthesis of chiral biaryls.

The first part dealt with the kinetic resolution of racemic biaryl using MBH carbonate as an electrophilic partner. The racemic starting materials were synthesized using the methodology developed by Kürti, while the kinetic resolution is the extension of Zhao's earlier work on the atropenantioselective acylation of phenols. Although the selectivity factor remained moderate to good, the starting material could be recovered with excellent enantioselectivity at the expense of the yield. The second part concerned the catalytic enantioselective N-alkylation of sulfonamides. This unprecedented atropenantioselective process provided the novel type of axially chiral tertiary N-aryl sulfonamides in excellent yields and enantioselectivities.

The work reported in this paper provided a new practical method for the synthesis of NOBIN derivatives and might help the future exploitation of this type of ligands, known for a while but still underexploited due to their inaccessibility. In view of the importance of sulfonamides in synthetic and medical chemistry, the easy access to axially chiral tertiary N-aryl sulfonamides would certainly attract the attention of wide readerships of this journal. The transfer of axial to point chiral chirality described in the conversion of 9ba to 11 (scheme 5) by reductive Heck is of interests. In fact, highly catalytic enantioselective reductive Heck is still difficult to accomplish. For a recent example dealing with the synthesis of enantioenriched oxindole related to 11, see: *ACIE*, 2017, 56, 3987.

Although I found that the introduction on the axially chiral amino alcohol NOBIN is a bit too long, overall, the paper is well-written and the supporting information is also of high quality. I highly recommend its publication in *Nat Commun*.

Reviewer #1:

(1) The only minor change to be suggested is to provide the configuration of indoles **11** and **12**.

Response: The absolute configuration of indoles **11** and **12** were assigned by the conversion of **11** to **13** followed by single crystal X-ray analysis. This was included in the updated Scheme 5 and in the SI.

Reviewer #2:

(1) The current study used acid-catalyzed reaction to prepare racemic NOBINs, which were then used for resolution. However, the chiral acid-catalyzed asymmetric synthesis of NOBINs has been reported (Tan, B. et al. *Angew. Chem. Int. Ed.* **2017**, *56*, 16308), which seems to be more practical and attractive. In this case, the authors may need to clearly state the advantages of the current method over that report.

Response: we appreciate the comment and certainly agree that work from the Tan group represents a significant breakthrough in enantioselective NOBIN synthesis. Nonetheless, we included in the discussion part that there is a limitation on the Tan method to require an aryl protecting group on the substrate to get high ee (>90%). The deprotection of that would be rather troublesome. Our method can produce analogous NOBIN derivatives with a practical procedure, despite the inherent limitation of kinetic resolution.

(2) The authors previously reported a kinetic resolution of NOBINs through NHC-catalyzed atroposelective acylation (*Angew. Chem. Int. Ed.* 2014, *53*, 11041), and the strategy gave NOBINs with excellent enantioselectivities. The author mentioned they tried the previous conditions for the substrates in the current study, but no data was reported in the manuscript or in the SI. Is the previous strategy compatible with the current substrates?

Response: The NHC-catalyzed procedure was not successful for the Ms-protected substrates that are easily accessible in this study ($S < 5$). The representative data was included in the updated supporting information as part 3.

(3) In the abstract, the authors claimed the importance of axially chiral N-aryl sulfonamides in ‘drug delivery’, however, the example provided by the authors in the main text is used as a NMDA antagonist, which seems to be a drug, rather than used for drug delivery.

Response: we truly appreciate this comment. We have changed the description of it to “great potential in drug molecules”.

(4) The structures of the amine catalysts in Table 1 are not in a consistent format. The authors should also check the stereocenters in the catalyst, e.g. the absolute configuration of chiral centers in (DHQD)₂PHAL, (DHQD)₂AQN, and (DHQD)₂Pyr should be the same.

Response: We appreciate this correction. We have updated this scheme to keep the absolute configuration of chiral centers in (DHQD)₂PHAL, (DHQD)₂AQN, and (DHQD)₂Pyr as well as their conformation identical.

(5) In Scheme 5, what is the ee of compound **12**? There is an optical rotation data for chiral **12** in the supporting information, so the author should also report the ee value and add the HPLC chart.

Response: The ee of **12** was determined to be the same as **11**. The scheme has been updated. The HPLC chart of compound **12** was also added in SI.

(6) The authors mentioned some of compounds **4** exist as a mixture of two isomers due to the presence of two axially chiral bonds. In this case, there should be four peaks for racemic **4** in the HPLC charts. However, there are only two peaks in all the HPLC charts of compounds **4**. What’s the reason?

Response: The use of different catalyst led to the formation of **4** in different dr. Single diastereomer was obtained in the racemic reaction, so only two peaks were observed in the HPLC charts of racemic compound **4**. In contrast, two diastereomers were obtained in the enantioselective reactions (the ethyl MBH carbonate reacted with Ts protection NOBIN derivative). Single diastereomer was also obtained in the tertbutyl MBH carbonate reaction.

The HPLC charts of **4ha** in different catalysts:

Reviewer #3:

(1) The transfer of axial to point chiral chirality described in the conversion of **9ba** to **11** (scheme 5) by reductive Heck is of interests. In fact, highly catalytic enantioselective reductive Heck is still difficult to accomplish. For a recent example dealing with the synthesis of enantioenriched oxindole related to **11**, see: *ACIE*, 2017, 56, 3987.

Response: We have had a brief discussion and included this important indoline synthesis as ref 34. In our studies, however, the conditions listed in our manuscript worked out more efficiently.

(2) Although I found that the introduction on the axially chiral amino alcohol NOBIN is a bit too long, overall, the paper is well-written and the supporting information is also of high quality. I highly recommend its publication in *Nat Commun*.

Response: we appreciate the suggestion. We have reduced the introduction on NOBIN part to keep our manuscript more concise.

REVIEWERS' COMMENTS:

Reviewer #2 (Remarks to the Author):

The authors have addressed all the issues raised by the reviewer thoroughly. Additional experimental results, detailed explanations, and revised structures have improved the quality of the manuscript. This reviewer recommend its publication in Nat Commun.